# Thermogravimetry Applicability in Compost and Composting Research: A Review

Manuel Jesús Díaz [1,]*, Mercedes Ruiz-Montoya [1], Alberto Palma [1] and M.-Violante de-Paz [2]

1 PRO2TECS. ETSI. Universidad de Huelva, 21071 Huelva, Spain; mmontoya@uhu.es (M.R.-M.); alberto.palma@diq.uhu.es (A.P.)
2 Dpto. Química Orgánica y Farmacéutica, Facultad de Farmacia, Universidad de Sevilla, 41012 Seville, Spain; vdepaz@us.es
* Correspondence: dblanco@uhu.es

**Featured Application: Thermogravimetric analysis (TGA) allows one to simulate the conditions of thermal degradation under different conditions (pyrolysis, combustion, etc.) and can be applicable to compost samples. Different physical and chemical properties of composts (main compounds, thermostability, etc.) can be analyzed by this technique.**

**Abstract:** Composting could be a suitable solution to the correct treatment and hygienization of several organic waste, producing compost that can be used in agriculture. The evolution and maturity of this process has been studied using a variety of techniques. One very promising technique for these studies is thermogravimetric analysis. On the other hand, the compost can be used for a variety of purposes different to the agricultural one, such as direct energy by combustion or energy and products by pyrolysis and its suitability can be measured by thermogravimetric techniques. With these goals, a bibliographic analysis has been done, applying Preferred Reporting Items for Systematic Reviews and Meta-Analyses PRISMA methodology, to the use of thermogravimetric equipment applied to the study of composting and compost uses. According to the methodology for PRISMA systematic reviews, the following databases have been searched Google Scholar, Web of Science, Mendeley, Microsoft Academic, World Wide Science, Science Direct, IEEE Xplore, Springer Link, Scopus, and PubMed by using the terms "thermogravimetry AND (compost OR composting) AND NOT plastic".

**Keywords:** compost; maturity; thermogravimetric analysis

## 1. Introduction

The increase in organic waste, whether it comes from food, from the agro-food industry or from forestry waste, is a serious problem due to the consequences that its inadequate management may cause [1]. Composting could be the most suitable solution to the correct treatment and hygienization of organic waste, producing compost that can be used in agriculture. In this sense, the applicability in terms of economy and efficiency of composting has been demonstrated [2]. It has been amply demonstrated that the application of mature compost, in the adequate doses, increases crop yields significantly to different crops. The crops in which the application of compost is possible are very varied, depending on the maturity, granulometry, moisture, presence of impurities, etc. [3]. In addition, new uses are being developed for compost such as its use as an energy source through combustion, gasification or pyrolysis due to the high diversity of compost and qualities that can be offered.

On the other hand, thermogravimetric analysis (TGA) allows one to quantify the change in mass weight under a selected heating rate and controlled atmosphere. Both weight changes and temperatures in the resulting thermogram can be studied. Among the advantages of using the TGA [4] that it requires very little sample quantity, (10–100 mg)

offers reproducible and very accurate results and are obtained in a short time (depending on the temperature to be reached and the heating rate selected could be highlighted). In addition, due to the low cost of the analyses as well as the speed in the results, it allows one to increase the number of analyzed samples and, therefore, to diminish the implicit heterogeneity in the compost samples. Moreover, thermogravimetric methods could be applied to compost samples, without pre-treatment, and this allows them to be used as routine methods to supervise the evolution of the composting process [5]. Schnitzer and Hoffman [6,7] and Mitchell and Birnie (1970) [8] proposed the use of thermal analysis for the characterization of soil and compost humic substances. In this form, thermogravimetric techniques could be used to clarify the degradability of organic matter and to evaluate qualitatively the stabilization rate of this matter, as well as the recalcitrant C in compost [9,10]. Several authors such as Dell′Abate et al. [11], Dell′Abate et al. [12], Melis and Castaldi [13], Lyons et al. [14], Muñoz et al. [15] and Onwosi et al. [16] identify thermogravimetric techniques among the useful methods for the identification of compost stabilization.

TGA is a technique in which the change in weight (either its increase or decrease) during the heating of the sample is measured. This heating is usually fixed. The first derivative of the weight change with respect to time (DTG) allows a better view of the points (temperatures) at which the greatest speeds of weight change occur. DTG is a more sensitive parameter compared to TGA and it is usually related to the reactions that occur at that temperature. When the temperature at which the weight loss occurs is higher, the organic fraction that is degrading is consequently more resistant and structurally ordered. The identification at which the mass loss is maximum, as well as the overlapping transformations, are clearly shown as peaks in the DTG plot.

An additional advantage of these techniques is that they are very synergistic, and commonly used, with other techniques such as calorimetric [Differential Scanning Calorimetry (DSC), Differential Thermal Analysis (DTA)] and analytical (GC-MS, FTIR) techniques.

Thermal analysis can give us information on data such as moisture content, certain compounds evolution and behavior of the material in relation to temperature changes among others and to complement chemical characterization data obtained from compost could be used. Moreover, it can be used for the improvement and quality control of the compost production processes.

The objective of this study is to carry out a bibliographic review of the uses and benefits that thermogravimetric analysis can offer in the study of both the quality of compost and the composting process.

## 2. Materials and Methods

The Preferred Reporting Items for Systematic Reviews and Meta-Analyses (PRISMA) methodology [17] has been used for our review. In this review, studies that include research on compost, specific compounds, composting evolution and different uses of compost or derivatives have been included. Data on the use of compost in plastic degradation experiences have been excluded.

Database searches such as Google Scholar, Web of Science, Mendeley, Microsoft Academic, World Wide Science, Science Direct, IEEE Xplore, Springer Link, Scopus, PubMed for publications in English by using the terms "thermogravimetry AND (compost OR composting) AND NOT plastic", have been carried out. In a first step, abstracts of the publications were analyzed in order to identify the publications to be reviewed with their full text. In a second step, for the selected articles, a detailed reading of the text and its conclusions has been carried out.

## 3. Results and Discussion

*3.1. Composting Evolution Research by Using a Variety of Techniques.*

3.1.1. Composting Evolution Research by Thermogravimetry

Various authors have published results derived from the use of thermogravimetric techniques in the analysis of compost and its evolution. These techniques have been also

used to check the process of maturation or stabilization of the compost (Section 3.2). Thermal studies in compost are directly related to the degradation of hemicelluloses, cellulose and lignin, since composts are mainly formed by these compounds, or to the humic and fulvic acids formed during composting.

Different researchers [18–24] exposed three main phases in the thermal degradation of compost: (i) the first phase is between an initial and final temperature of 25 and 120 °C; (ii) the second phase is between 200 °C and 350 °C; (iii) the third phase is between 350 °C and 700 °C (Figure 1).

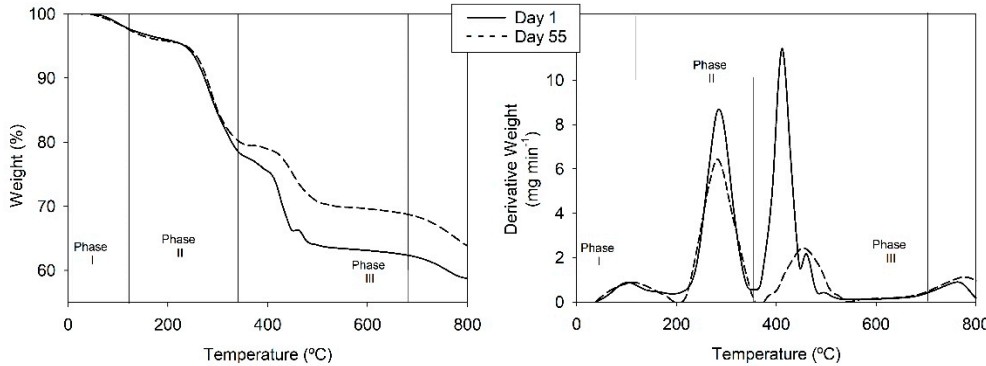

**Figure 1.** TGA [weight and DTG (derivative weight)] corresponding to the Municipal Solid Waste (MSW) evolution in composting (figures shown are from the authors).

In the first phase, first peak at DTG (<200 °C), this water content reduction is due to a thermal dehydration process of the compost samples [25]. Other authors also indicate a decarboxylation in labile compounds [26].

In the second phase (200 to 350 °C) the thermal decomposition of the organic components already takes place. In this sense, between 200–300 °C (second peak at the DTG) the thermal degradation of simple organic matter and semi-volatile compounds such as hemicelluloses, cellulose, degradation of aliphatic structures and microbial cell walls can be associated [27]. It has been proven that as composting time increases, weight losses in the range of thermal degradation of carbohydrates are constantly decreasing [5,11,13,28–33]. Moreover, Sharma et al. [34] identified the breakage of polysaccharide and cellulose fibers in this interval. As composting progresses, lower peaks of DTG in this area, due to the decreased concentration of easily degradable compounds, are noted. Conversely, Blanco and Almendros [35] indicated that carbohydrates were probably responsible for some of the peaks found between 300–315 °C.

In the third phase, under higher temperatures (350–700 °C, third peak in the DTG) degradation of aromatic structures and organic polymers such as lignin, or others synthesized during the compost stabilization process have been associated [36]. Dhyani et al. [23] found that dietary fibers and lignin break down over a wide range of temperatures and that they are the main components that break down beyond 400 °C. DTG peaks at this area increase with the days of composting. This trend towards a progressive transformation of readily biodegradable compounds into macromolecules is being suggested. These formed compounds are known as humic-type substances [5,26,37] also indicated a shift towards higher temperature, more thermally stable, the DTG peaks in the 350–700 °C zone as composting time increases [13,38–40]. This shift to higher molecular complexity and aromaticity during the composting process is attributed [9]. Conversely, Baigorri et al. [27], by combining TGA and FTIR, proved that between 380–460 °C the decarboxylation phase is lower for gray humic acid compared to brown humic acid. Furthermore, Dhyani et al. [23] showed peaks of DTG after 600 °C attributed to the volatilization of carbon in the compost.

### 3.1.2. Composting Evolution Research by Thermogravimetry and Thermal Analysis

Other authors have used thermogravimetric analyses combined with Differential Scanning Calorimetry (DSC) by which, in addition to changes in mass, endothermic or exothermic peaks —attributed to evaporations, transitions or reactions occurring in the samples— can be shown. In this sense, endothermic peaks at <200 °C due to dehydration and between 600–700 °C attributed to thermal degradation of carbonates can be found [41]. On the other hand, two exothermic peaks at the range of 200–700 °C corresponding to the degradation of organic matter are also found in compost. The first exothermic peak (200–350 °C, part of phase II) corresponds to the combustion of carbohydrates such as cellulose and lignocelluloses and the second exothermic peak (350–700 °C, part of phase III) corresponds to the thermal degradation of more complex aromatic structures [42].

Dell'Abate et al. [11], Otero et al. [42], Bernabé et al. [43]; Martín-Mata et al. [44] and Pelegrín et al. [45] proposed the joint use of thermogravimetric analysis (TGA) and differential thermal analysis (DTA) as a technique to evaluate the grade of stabilization of composts. R1 (thermostability index) is the ratio between the weight losses associated with the second (200–350 °C) and third phase (350–700 °C) of compost thermal degradation. The relative amount of the thermally less stable organic fraction in relation to the one more stable is evaluated by this parameter. The justification for their use is that both the mineralization and, during the biological stabilization process, the conversion of organic components into humic substances and the energetic changes produced by the metabolic processes that take place during the compost maturation process are closely related. The authors mentioned above indicated that the R1 ratio should increase during composting. In this sense, Gómez et al. [38] demonstrated that the composting process, during the maturation phase, increases the degree of aromaticity in the resulting compounds and the results of the aromaticity index are in accordance with the thermal profiles obtained in the samples. Bernabé et al. [46] showed, comparing the TGA curves of the compost samples, that an increase in mass that degrades above 700 °C occurs as the composting time increases. Moreover, Dell'Abate and Tittarelli [47] showed that organic matter stabilization had been associated with an increase of water retention, and Mondini et al. [48] demonstrated that the parameter R1 with both humification index and degree of humification are significantly correlated during composting.

The use of the TGA and the R1 parameter indicates the high sensitivity of this technique and parameter to changes induced by the composting process. However, Fernández et al. [49] concluded that it is necessary to test the observed trend with composts of different nature in order to analyze the relationship between thermal and biological stability. If this relationship can be found, DSC-TG thermal analysis could be taken into account among the usual methods in the characterization of composts.

### 3.1.3. Composting Evolution Research Studies by Combining Different Thermal and Analytical Techniques

In addition, other authors' several combined techniques in compost characterization have been studied. In this sense, Carballo et al. [32], Bernabé et al. [43] and Ouaqoudi et al. [50,51] combined the traditional physicochemical parameters, Fourier transform infrared spectroscopy (FTIR), TGA and differential thermal analysis (DSC), to determine the maturity of lignocelluloses compost. The suitable correlation of these techniques in the thermal behavior of compost and its evolution (mainly by humic and fulvic acids evolution) has been demonstrated [44,52]. Moreover, Qu et al. [53] have been integrated into their study the use of FTIR, TGA, differential scanning calorimetry and scanning electron microscopy. In addition, Hussain et al. [54], by the combined use of FTIR, UV-vis, DTA-TGA and SEM, provide substantiating evidence on the evolution of composting by combining different complementary techniques. Moreover, Ouaqoudi et al. [55] demonstrated that the combined use of TGA with TMAH-thermochemolysis and FTIR in the study and characterization of the chemical structure of humic fractions in composts provides consistent and complementary results. Lastly, Smith et al. [24] employed TGA,

FTIR and $^{13}$C NMR to prove the degradation of both labile organic matter and complex aromatic and lignin compounds during composting.

### 3.2. Compost Maturity Evaluation Research by Using a Variety of Techniques

In general, the term maturity, in composting, with the degree of microbial activity, to the presence/absence of volatile compounds or consumption of oxygen has been related. Various parameters (temperature, BOD, $O_2$, $CO_2$, humic and fulvic acids, etc.) and methods as maturity parameters of the composting process have been developed [56]. However, there is not, to date, a unified method suitable for use for composts from various origins, due to the different physicochemical characteristics of the organic waste that can be composted. Among the most widely recognized methods are the respirometric methods [57].

### 3.2.1. Composting Maturity Research by Thermogravimetry

Different authors have proposed thermogravimetric methods (sometimes combined with other methods) due to the great advantages that these methods represent both in terms of speed and the no-need to pre-treat of the sample [9,11,33,36,42,58,59]. Moreover, Sharma et al. [34] have been able to differentiate the efficacy of different composting methods through these techniques and several authors have reported that this behavior is consistent with the fertilizer stabilization process [11,13,33,38,60].

### 3.2.2. Composting Maturity Research by Thermogravimetry and Thermal Analysis

The compost maturity could be characterized by thermogravimetric and thermal techniques or differential scanning calorimetry (DSC) and thermogravimetric analyses (TGA) [5,12,13,37,38,42, 48]. In this sense, during the maturation process, the solid content and the microorganism populations are reduced. The trends followed by the degradation profiles obtained from DTG-DTA during this maturation are consistent with this reduction [42]. Moreover, different authors pointed out that the evaluation of the maturity degree of compost through thermogravimetric data (TGA+DSC) was useful and coherent with the quantitative information coming from the chemical analysis of the humidified fraction of organic matter in compost [11,58,59]. This information, through the parameter R1, allows us to quantitatively evaluate the stabilization of organic matter in the composting process and could be considered as a reliable parameter to check the maturity level of organic matter [12,48]. In this form, different studies have been published in which compost with R1 $\geq$ 0.8 can be considered mature [33,59]. On the contrary, Dell'Abate et al. [5] stated that the amount of easily biodegradable organic matter in compost is not directly measured by R1; therefore, it does not correctly evaluate the biological stability. In this sense, Ouaqoudi et al. [51] reported that although different composts gave the same final value of R1, they showed different values in other maturity parameters evaluated. Baffi et al. [61] stated that no significant correlation had been observed among dynamic respiration index (DRI), humification indices and R1.

In another paper, Blanco and Almendros [35] showed that DTG peaks corresponding to the degradation of more refractory compounds such as lignin and other resistant aromatic structures increases with the maturity of the compost. In addition, they indicated the displacement of the DTG peaks as the compost matures. These effects, with both increased aromaticity of the final products and increased oxygen content in the functional groups of the more easily degradable compounds, were both associated.

### 3.2.3. Composting Maturity Research by Combining Different Thermal and Analytical Techniques

Using different technical associations to analyze the compost maturity have been proposed. In that form, Ouaqoudi et al. [51] combined the traditional physicochemical parameters, FTIR and GC-MS spectra and TG/DTG-DTA curves; this way, it was possible to verify that the final product remains primarily as a lignocellulosic macromolecule. In addition, Lim et al. [62], analyzed the maturity and stability of vermicompost by using FTIR and UV-vis data combined with TGA concluding that spectroscopic techniques together

with TGA are promising approaches to characterize the evolution of the composting process. In addition, Morales et al. [63], by using advanced instrumental techniques, such as TG/DTG and EEM fluorescence spectroscopy, have demonstrated the applicability of that combination to compost maturity.Srivastava et al. [64] concluded in the same way by using the combination of FTIR, TGA and DSC.

### 3.3. Thermogravimetry in Compost Combustion

Compost treatments generally result in variations not only in the water content but also, and more importantly, in the fuel characteristics of the starting materials [65]. During composting, the overall energy content of the composted waste is reduced as a result of the partial decomposition of the initial organic matter and a concomitant detrimental effect is also observed due to the increase in ash content [66]. However, compost still preserves most of its calorific power, which makes it possible to produce energetic compost either for direct combustion, gasification or pyrolysis. For this reason, and from an energy point of view, compost can be considered a potential biomass fuel [67]. General changes that occur during the thermal behavior of composting have been evidenced by TGA: (a) enrichment of non-volatile high molecular weight compounds, (b) synthesis of microbial biomass and (c) chemical transformations such as humic substances formation. Microbes convert 60% to 70% of carbon to $CO_2$ and utilize the remaining 30% to 40% in their body as cellular components [68]. During degradation of organic compounds, losses in calorific value are observed due to the bio oxidation of organic substances

Malat'ák et al. [69] experimentally determined the energy potential of briquettes prepared from an oversized fraction of wasted wood compost and the theoretical combustion characteristics compared to conventional fuel: wood logs. The most decisive parameter for energy use of selected samples of compost briquettes and spruce logs is the net calorific value (NCV). NCV greatly depends on water and ash contents in the fuel, both with a negative impact on the final NCV [70]. The authors analyzed non-combustible substances in that compost, total water content and volatile matter by TGA. These results, together with others obtained (stoichiometric calculations of individual samples, elemental analysis and gross calorific value > 16.0 MJ kg$^{-1}$), demonstrated that the compost under study possessed suitable properties for further use as an energy source. However, TGA results showed several times higher ash content in the samples of compost briquettes in comparison with woody biomass. Moreover, Malat'ák et al. [71] investigated about energy utilization of another type of untreated compost as a fuel (material inputs: cattle manure, sewage sludge and plant-based wastes such as leaves, wood chips, straw, hay, sawdust, and spoiled vegetables and fruits). Water and non-volatile components (i.e., ash content) were determined using thermogravimetric methods. The low calorific figures associated with this compost (<9.0 MJ kg$^{-1}$ on dry basis) were due in part to the enormous percentage of ash found (>60% on dry basis) and, therefore, this material could not be used as fuel without a pretreatment to reduce the incombustible content. To consider this biomass as energy compost biomass, whether for direct combustion or for its thermochemical conversion (pyrolysis), it was necessary to mix it with wood biomass (it was mixed with spruce chips in a 1 to 1 ratio). The most relevant experimental difference found between the combustion of compost briquettes and that of the prepared mixture was the flue gas temperature, which experienced an increase of almost 100 °C due to the net calorific value of the added wood biomass. However, with increasing excess air coefficients, the CO and $NO_x$ emission levels have found increased [69].

Another main raw material for composting is municipal solid waste (MSW). MSW is the result of human activity and, therefore, is made up of various types of waste materials that behave differently during thermal degradation processes (pyrolysis or combustion). In general terms, the thermal degradation of its individual components has been studied separately, which has masked the synergies that may take place during combustion and pyrolysis. Even though these synergistic interactions can arise between the components of the MSW during their thermal conversion and, hence, altering the degradation patterns

expected according to its composition, few studies have focused on discovering them. Gunasee et al. [72] approached this study by determining the apparent activation energy that occurs during the thermal conversion of MSW. In this sense, TGA and MS worked in tandem in the quantification of the synergistic mechanisms and associated thresholds, through the findings of the activation energies, the comparative studies of weight losses found experimentally with those calculated according to the initial components and the relative areas in the mass spectra.

In addition, a material that could potentially be used as a product with added bioenergy value through composting is waste derived from animal breeding. Singh et al. [73] investigated the potential use of chicken manure (fresh and composted) because of its high protein and carbohydrates contents due to the widespread practice of overfeeding chicken so that they reach the minimum body weight in the shortest period possible. However, from the TGA experiments (moisture content: 24% and 27% for fresh and composted manure, respectively; residual mass: 24 and 26% for fresh and composted manure, respectively) and stored energy data (6905 J g$^{-1}$ and 8239 J g$^{-1}$ for fresh and composted manure, respectively), it can be concluded that neither fresh nor mature manure compost are suitable materials for use as biofuel directly without additional modifications.

Thermogravimetry in Kinetic of Compost Combustion

López-Gonzalez et al. [74] investigated the potential valorization of compost from animal breeding by combustion and pyrolysis using thermogravimetric analysis (TGA) coupled with mass spectrometry (MS) detector in order to define its potential energy production. The kinetic parameters of the pyrolysis and combustion procedures were attained by TGA. They found a significant water rate in the samples studied, a fact that constitutes a serious drawback since it can cause feeding and fluidization problems. For this reason, it is recommended to conduct a pre-drying process of the material before its use as energy compost, a technique that usually involves high operating costs.

Thermogravimetric analyses (TGA) is a suitable method to determine the kinetic parameters of thermal degradation processes in composts [74–78]. In this sense, the assessment of kinetic parameters can be carried out from TGA data to which techniques known as model-fitting and model-free are applied, including the identification of the kinetic reaction model or lack thereof. In general, the kinetic model used was derived from the Arrhenius equation and first order reaction has been assumed [76], although some authors have described it as a function of other reaction orders that implies the existence of parallel reactions between the main components of biomass (hemicellulose, cellulose, and lignin) [77]. Based on TGA data, the activation energy (Ea) values involved in the composting processes could be calculated to find the maximum energy efficiency, i.e., the most profitable operating conditions for the production of an incinerable product. To determine that activation energy for the obtained composts, model-free methods (for example, the isoconversional method) are believed to be one-step kinetics and this is the reason why their use is suggested in determining combustion kinetic data. Of these isoconversional practices, the Flynn–Wall–Ozawa integral method may be one of the most widely employed [78,79]. In this line, Doña-Grimaldi et al. [77] investigated the valorization under combustion of MSW compost by TGA. They measured the maturation of MSW compost and studied the influence of the selected composting parameters, such as aeration and moisture content, on the actual heat capacity of the material. Based on the thermal degradation (TGA) and differential mass loss (DTG) thermograms they observed that both composting operating parameters affected the biodegradability of compost (Kb = 0.36 and 0.42) and more importantly, their gross heating values. An evident growth in heating values was observed for both reactors after 40 days (when the maturation stage had finished). The increments were linked to fulvic and humic acids formation since, according to Kastanaki et al. [80], the calorific values of these acids are analogous to those found for lignite (coal). It was concluded that composting MSW under appropriate conditions was not only an ideal process, able to concentrate and stabilize the calorific power of the

material, but also a convenient body drying process. Raclavska et al. [81] demonstrated that the utilization of MSW based compost for energy generation is possible when some technological modifications of the materials are made. The aim of these adjustments is to produce final fuels with gross calorific figures above 10 MJ kg$^{-1}$ based on dry matter. When aiming to use the compost as low-energy fuel, the main problem they found, which is the foremost drawback experienced by other authors, was its excessive moisture content. The authors observed that, to generate a net calorific value (NCV) greater than 10,000 kJ kg$^{-1}$ a reduction of the water content of the compost to values close to 12% was necessary. This reduction in moisture could take place spontaneously by means of extending up to six months the maturation periods or by wastepaper addition. The experiments demonstrated that the net calorific value of fuel could be readily optimized for mature compost (>11,000 kJ kg$^{-1}$ dry matter) without pre-drying the compost by adding cardboard. The energy content of the material was significantly optimized when cardboard was incorporated at the end of the composting process. Synergistic effects were related to the dependency of the effective activation energy versus the conversion. To obtain the dependency relationship between the activation energy and the degree of conversion the Friedman method was chosen, which was adequately described by Aboyade et al. [82].

The pyrolysis and combustion of sewage sludge (fresh and composted) and the kinetics of the different steps involved in have been investigated by García Barneto et al. [83]. Although the composition of sewage sludge is characterized by six groups of components [84], it is necessary to know the sludge fractions according to temperature response, rather than chemical composition. For this reason, the pyrolysis processes of both fresh and composted sewage sludge were mimicked considering the five portions present in raw sludge: (a) organic compounds of low thermal stability; (b) hemicellulose; (c) cellulose; (d) lignin and plastic; and (e) inorganic. Being the thermal behavior of the main components of the sludge were studied, it was found that the thermal degradation of each fraction followed an n$^{th}$-order kinetic. In addition, both their kinetic parameters (pre-exponential factor and apparent activation energy) and their thermal behaviors were equivalent to those found for the main components of lignocellulosic biomass, i.e., hemicellulose, cellulose and lignin. In addition, the kinetic parameters correlated experimental and calculated data for sewage sludge for all heating rates with remarkable accuracy. The methodology used represents an effective means to differentiate between biodegradable and non-biodegradable matter.

*3.4. Thermogravimetry in Compost Pyrolysis*

The pyrolysis of the compost samples was studied by TGA. Pyrolysis of diverse compost from different raw materials has been studied with different objectives in the last years. The main weight loss was observed between 209 and 373 °C by several authors. This range is narrower than that found by other authors in the pyrolysis of lignocellulosic biomass for which the highest weight loss is normally found between 200 and 450 °C [75]. Differences in lignin content may explain this discrepancy because lignin is the thermally most important material among the components of biomass and compost. In addition, pyrolytic carbon is mainly derived from lignin [85]. On the other hand, other authors point to the high ash content of the compost as being responsible for the difference in temperature ranges mentioned above [86]. Ryu et al. [87] investigated the technology of manufacturing pellet blends for energy production from two discarded materials in industry: coal tailings from coal cleaning processes and spent mushroom compost (SMC) a mixture of fibrous compost substrate and a wet casing layer used during mushroom production. Differential thermogram (DTG) of air-dried SMC and coal tailings demonstrated the differential behavior of the components. SMC materials under two temperature ranges (230–580 °C and 680–790 °C) have been degraded. Pyrolysis of the lignocellulosic components has been attributed to the first range. In addition, a peak corresponding to 310°C (DTG) to cellulose decomposition is usually attributed. The second interval to CaCO$_3$ decomposition is normally attributed although also carbonaceous residues are responsible for the degradation

suffered between 350–670 °C with a large mass loss peak. Thermal degradation of these components at 670 °C is assumed to be complete.

### 3.4.1. Thermogravimetry in Compost Bio-Oil Production

One of the main studied objectives has been the bio-oil production through pyrolysis. Finney et al. [88] and Garrido et al. [89] compared different heat treatment technologies (pyrolysis and combustion) to determine an appropriate method of energy recovery from mushroom compost. In both cases under pyrolysis process a bio-oil with a low calorific value has been obtained. However, the moisture reduction and low ash content makes it a suitable raw material. Garrido et al. [89] also proved that pre-treatment processes (roasting) can alter the composition of the bio-oil produced.

### 3.4.2. Thermogravimetry in Compost Hydrogen Production

On the other hand, López-González et al. [74] studied the pyrolysis of compost to obtain hydrogen. A significant production of $H_2$ at temperatures higher than 300 °C has taken place. This was mainly due to dehydrogenation reactions of the compounds present in the compost. The high presence of alkaline and alkaline-earth metals apparently catalyzed the process since the hydrogen was obtained at lower temperatures than those observed in biomass (200–450 °C). In this sense, they demonstrated that a significant concentration of Ca could promote the production of $H_2$, while the presence of K has been associated with the production of $CH_4$. In this context, thermogravimetric analysis combined with mass spectrometry (TGA-MS) to study the thermal behavior of pyrolysis of compost and biomass mixtures has been studied [86,90,91]. The decomposition of the compost mixture took place mainly from 209 to 373 °C, with a total weight loss of 23%. In this case, the main gaseous products generated during pyrolysis were $H_2O$, CO and $CO_2$. For this reason, this study concluded that the pyrolysis of these composts did not seem to be a promising process due to the low values of superior calorific value and biochar. In another study, Lee et al. [92] and Giwa et al. [93] analyzed the thermal degradation behavior, calorific value and gas spectrometry during pyrolysis among compost from food waste. They concluded that these composts can be used as a promising alternative fuel. Moreover, when the compounds formed during pyrolysis (TGA-MS) have been studied, the maximum emission peak was obtained at temperatures around 350 °C for most compounds, being the most abundant CO, $H_2O$ and $CO_2$. It is remarkable the production of $H_2$ at high temperatures (>450 °C, peak extended up to 1000 °C). It was hypothesized, based on experimental data, that $H_2$ production took place through both the reverse of the methanation reaction and the steam reforming reaction [94,95]. The slow production rate of $H_2$ was correlated with the slow carbonization process of the char, evidencing char thermal cracking and dehydrogenation reactions. The presence of Ca in the residue content could promote the $H_2$ production via steam reforming reactions, whereas the presence of K was associated to the production of $CH_4$.

### 3.4.3. Thermogravimetry in Kinetic of Compost Pyrolysis

The kinetics of pyrolysis of composts have been studied by thermogravimetric analysis. In this sense, the first papers in the bibliography [96,97] related to the kinetics of composts pyrolysis, by using thermogravimetric analysis, are related to municipal solid waste composts. The kinetic model proposed by Sesták-Berggren [98] is shown to be adequate to describe the first step of thermal decomposition of those compost components, which is related to the decomposition of cellulosic products. On the other hand, Crespi et al. [96] and Silva et al. [97] calculated the kinetic parameters by using the Flynn–Wall—Ozawa isoconversional method at a 95% confidence level. Later, other kinetic models, based on the selective degradation, by thermogravimetric techniques, of the compost components during pyrolysis were also studied [98]. In addition, Giwa et al. [93] observed a certain catalytic role of composting by-products in the pyrolysis of food waste composting. In this sense, the thermal behavior of these wastes could be successfully modeled using first and

third order chemical reactions within the temperature ranges of 200–360 and 360–510 °C, respectively, obtaining activation energies varying between 25.68 and 41.89 kJ mol$^{-1}$.

Sait et al. [99] focused on the pyrolysis and combustion processes for date palm biomass wastes using TGA. This study gave them the opportunity to draw their attention to the kinetic analysis of this biomass and a detailed description of the methodology used is included in the article. Their calculations were based on the Arrhenius equation since, fundamentally, every kinetic model obeys it. The extent of the reaction was measured as a function of the mass loss in the biomass sample or the mass of volatile compounds generated. The activation energy and the pre-exponential factor of the Arrhenius equation for the different peaks found in the thermograms, both in combustion and in pyrolysis, have been obtained. The regression analysis yielded high correlation coefficients (0.81–0.99%) for the studied components. Different activation energies for various date palm biomass, as a result of different chemical reactions under thermal degradation have been found. Therefore, the activation energies under oxidative atmosphere appeared to be higher than under nitrogen atmosphere. From these data, the authors also concluded that thermal degradation of palm biomass, under oxidative atmosphere, had been influenced by heat and mass transfer limits. Due to the high calorific values and high volatile content of date palm seed and leaf, it was concluded that they can become useful sources of energy, chemicals and bio-char.

Palma et al. [100] evaluated the influence of municipal solid waste composting conditions on the pyrolysis susceptibility. The activation energy values of these composts are in the range of 57.78–581.69 kJ mol$^{-1}$. The assumption that the compost pyrolysis adjusts to a first order kinetics seems to be a suitable approach for municipal solid waste compost. Analysis of the gases produced by the pyrolysis process revealed that hydrogen increases in concentration as the composting time advances to an intermediate time. In this way, the composting process could be a suitable pre-treatment to improve the obtaining of hydrogen through pyrolysis. Moreover, Garcia Barneto et al. [83,101] studied the effect of composting on hydrogen production from biomass and compost pyrolysis. Based on the data shown, hydrogen production is positively affected by composting, increasing the hydrogen concentration in the raw gas from 15.2 to 22.6% in volume with respect to non-composted biomass. It is accompanied by a decrease in carbon monoxide concentration. This fact could be explained as a consequence of the physicochemical changes associated with composting, mainly in the thermophilic stage. After this stage, no progress on hydrogen production has been observed. Synergistic effects were related to the dependency of the effective activation energy versus the conversion. The activation energy-dependency curves versus the extent of conversion were obtained using the Friedman's method as described in detail by Aboyade et al. [82].

### 3.4.4. Thermogravimetry in Activated Carbon from Compost Pyrolysis

Activated carbon has also been obtained by pyrolysis, using compost as raw material. In this sense, Qian et al. [102] experimented with obtaining active carbon from cattle manure compost through chemical activation with zinc chloride. In this study, thermogravimetric analyses were used to monitor the pyrolysis process of compost. The activated carbons obtained showed high adsorption capacities.

### 4. Conclusions

It is clear that thermogravimetric analysis provides important information for both the evaluation of the composting process and the maturity of compost obtained. These techniques give us information on the degree of humification of the organic matter. This information has been corroborated in different studies and with different analytical techniques. Moreover, the presence of physical processes and chemical reactions attributable to the thermal degradation both in an oxidizing atmosphere (combustion) and in an inert atmosphere (pyrolysis) and their products can be corroborated by thermogravimetric analysis techniques.

**Author Contributions:** Conceptualization, modeling and writing—original draft preparation M.J.D.; writing, methodology and validation, M.-V.d.-P., M.R.-M. and A.P.; funding acquisition, M.J.D. All authors have read and agreed to the published version of the manuscript.

**Funding:** This study received financial support from Regional Ministry of Innovation, Science and Enterprise, Government of the Junta de Andalucía (Operational Programme FEDER Andalusia 2014–2020. Project UHU-1255540), Spain.

**Institutional Review Board Statement:** Not applicable.

**Informed Consent Statement:** Not applicable.

**Data Availability Statement:** The showed data have been obtained from the mentioned databases and references.

**Conflicts of Interest:** The authors declare no conflict of interest according to Data Availability Statement ("MDPI Research Data Policies" at https://www.mdpi.com/ethics).

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
