# Peer review of "Thermogravimetry Applicability in Compost and Composting Research: A Review"

_applsci, doi:10.3390/app11041692_

Round 1

Reviewer 1 Report

Development and detailed investigation of methods to control the composting process can provide useful information for the science and for practice, as well. Therefore, review manuscript of applsci-1102765 can be considered as interesting for the readers of Applied Sciences. Thermogravimetric analysis has great potential to detect the chemical and biochemical changes during the different stage of composting process. Review study has made based on relevant references. manuscript contains obviously valuable and interesting results.

Comments, suggestions

However authors focus on the applicability of TGA for control of composting process but I suggest to summarize briefly the applicability of other method in Introduction section, as well.

I suggest the authors to present the literature data in tables and create/use figure to improve the visibility of the main findings and recommendations related to the application of TGA.

In my opinion the Abstract section is too general and with present content it is not summarize the main ‘essence’ of the study. I suggest the authors to highlight clearly the main applicability of TGA method in Abstract section.

Author Response

I am enclosing the new revised version of our article “Thermogravimetry Applicability in Compost and Composting Research: a Review”.

          In this new version, we have taken into account the comments of the others reviewers and your suggestions.

          And once again, we would like to thank  for your help and to express our readiness to take into account any further observations you may consider necessary.

Reviewer 2 Report

This is a mini review, carried out using the PRISMA methodology and regarding the benefits of TGA in studying both the quality of the compost and the composting process.

The paper shows a good number of current and related research and provides some conclusions of its own.
I believe that the paper is not excellent but it still deserves to be published after a more critical analysis, substantially highlighting something more especially on the kinetic data that can be obtained from the TGA analysis.
Some tables would help  for easier reading.
Pay attention to the numerous typos.

Author Response

(The authors gave the same response as above.)

Reviewer 3 Report

After reading the manuscript, I suggest that the paper under review should be rejected. The authors divided the TGA profiles into three stages: 25-125 ° C, 200-400 ° C, 400-700 ° C. In the next sections of the manuscript  Authors changes the ranges of the stages very freely. In line 136-137 the range of the third phase is indicated as 350-600 ° C. What range do these temperatures belong to, the second or the third? What happens in the temperature range from 125 ° C to 200 °? This range is neither the first phase nor the second phase.

Due of these fundamental errors, the manuscript should be rejected. 

Author Response

(The authors gave the same response as above.)

Reviewer 4 Report

The manuscript entitled "Thermogravimetry Applicability in Compost and Composting Research: a Review", reports a review work on the use of thermo-gravimetry for the characterization of compost. The work takes into consideration 102 references of which the oldest dates back to 1965 and the most recent to 2020. There are seven articles of 2020. The subject matter is consistent with the purpose of the newspaper as well as being interesting and useful for those who work in the composting sector. It is clear that there has been a copious collection of information through the consultation of many articles. However, I believe that in the current version the manuscript is heavy to read and probably should be organized differently. Probably a different and more detailed subdivision of the information into several paragraphs, perhaps distinguishing the most important compounds, would make it easier to read. Furthermore, I note the complete absence of images in the manuscript. Bringing more images, for example of representative thermal curves, would be useful and more complete. In light of the above, I believe that the manuscript should be subjected to revision before being considered for its publication, especially with regard to the exhibition organization. Here are some suggestions for authors.

Minor suggestions: Abstract (Lines 19-20): The following sentence is unclear “in English by using the terms“ thermogravimetry AND (compost OR composting) AND NOT plastic ”.

(Line 64): The following sentence should be better explained “The characterization of products of thermal analysis can give us information… .. chemical composition”. It can hardly give precise and complete information on its chemical composition but only on a part of these.

(Line 72): Although the reference note is present, I believe more information needs to be given in the text regarding the PRISMA methodology.

(Lines 92-94): The text states “(i) the first phase are between an initial and final temperature of 25 and 120ºC; (ii) the second phase are between 200 °C and 400 °C; (iii) the third phase are between 400 °C and 700 °C. ". Perhaps it is more correct to indicate that the first phases are between 25-200 °C, in order to have no temperature ranges not considered and subsequently specify this temperature range in detail.

(Lines 95-103): This part would be easier and more useful if accompanied by representative images of TG and DTG thermigravimentric curves. There are no images in the text.

Paragraph 3.1. This paragraph talks about the DSC and DTA, but no mention has been made in the introduction where only the TG and DTG are mentioned. Yet these analyzes are usually done simultaneously.

Maior suggestion: The manuscript should be improved in its exhibition organization. In the current version it is hard to read and follow, not even being supported by images. A greater division into paragraphs, perhaps by the nature of the compounds considered, could make it more usable.

Author Response

(The authors gave the same response as above.)

Round 2

Reviewer 1 Report

Manuscript applsci-1102765 has an interesting topic which has relevance for the practice, as well. Authors have revised the manuscript significantly. Rephrasing, amendments and further information added made the manuscript more clear and complete. I accept authors’ reply and all modifications and recommend the manuscript for publishing.

Author Response

In this new revised version, we have taken into account the comments of the other reviewers and your suggestions. And once again, we would like to thank your for your help and to express our readiness to take into account any further observations you may consider necessary.

Reviewer 2 Report

OK, although the authors did not respond adequately to my observations

Author Response

Dear Reviewer: In this new revised version, we have taken into account the comments of the other reviewers and your suggestions. And once again, we would like to thank your for your help and to express our readiness to take into account any further observations you may consider necessary.

Reviewer 3 Report

Review report.

The work is entitled Thermogravimetry Applicability in Compost and Composting Research: a Review. Dear Authors, I would like to thank you for correcting and well organizing the information in the manuscript. The reviewed studies presented in the form of a review manuscript concern a very important issue, which is the assessment of compost maturity using the ratio of simple thermogravimetric methods. The ability to quickly assess the condition of the compost and correctly interpret the results is important in the case of composts consisting of different components. However, I have a few questions and doubts that arose while reading the manuscript:

Line 30- 32: 'Composting could be the most suitable solution…'. Why, according to the authors of the manuscript, is composting the most appropriate for the proper processing of organic waste? Please provide a brief justification. It may be more convenient to pyrolyze them directly or carry out the organic waste to a high temperature drying process (torrefaction process). Back then they were bacteriologically safe. In my opinion, composting is simply the cheapest way to treat organic waste.

Often when reading review works, I wonder what new information is presented in the work or what is the originality of this review article?

The authors themselves mention that in many publications you can find information about using gravimetric techniques for the analysis of compost. What information is missing from previous publications? I would ask for a brief supplement information what is novelties are presented in the manuscript?

Regards

Reviewer

Author Response

(The authors gave the same response as above.)

Reviewer 4 Report

The manuscript has been improved although there are still some inaccuracies listed below:

   (Lines 91-92). Paragraph titles are merged.

(Line 173) The paragraph title is not detached from the text.

The titles of the following paragraphs are the same:

3.4.3 Thermogravimetry in the kinetics of compost pyrolysis

3.4.4 Thermogravimetry in the kinetics of compost pyrolysis

Author Response

(The authors gave the same response as above.)
